# Dielectric Characterization of Non-Conductive Fabrics for Temperature Sensing through Resonating Antenna Structures

**DOI:** 10.3390/ma13061271

**Published:** 2020-03-11

**Authors:** Isidoro Ibanez-Labiano, Akram Alomainy

**Affiliations:** School of Electronic Engineering and Computer Science, Queen Mary University of London, London E1 4NS, UK; a.alomainy@qmul.ac.uk

**Keywords:** material characterization, smart clothing, temperature sensing, wearable technology

## Abstract

Seamless integration of electronics within clothing is key for further development of efficient and convenient wearable technologies. Therefore, the characterization of textile and fabric materials under environmental changes and other parametric variations is an important requirement. To our knowledge, this paper presents for the first time the evaluation of dielectric characterization over temperature for non-conductive textiles using resonating structures. The paper describes the effects of temperature variations on the dielectric properties of non-conductive fabrics and how this can be derived from the performance effects of a simple microstrip patch antenna. Organic cotton was chosen as the main substrate for this research due to its broad presence in daily clothing. A dedicated measurement setup is developed to allow reliable and repeatable measurements, isolating the textile samples from external factors. This work shows an approximately linear relation between temperature and textile’s dielectric constant, giving to fabric-based antennas temperature sensing properties with capability up to 1 degree Celsius at millimeter-wave frequencies.

## 1. Introduction

Dielectric characterization of materials is crucial to understand the iteration between electromagnetic waves with matters [1,2]. In order to fully develop textile-based technology, the dielectric properties of fabrics need to be quantified. These properties are affected by external factors, such as moisture or relative humidity (RH) content and temperature [3]. Previous works [4] have studied the impact of temperature on the dielectric properties of textiles, but without considering the experimental results on several scenarios. There are several cases where textile-based devices must withstand changes in temperature without changing its overall performance, so a deep understanding of the response under different temperature conditions is needed [5]. In addition, an understanding of this phenomenon can be used for sensing applications. Temperature is a crucial parameter to be measured in several fields and applications such as infrastructures, system maintenance, the food industry or body sensing [4,6,7]. 

Recently developed technologies within the metamaterials discipline have looked into this topic as well. Where the field theory is used to relate electromagnetic fields with parameters such as the permittivity (ε) and how these change as a function of temperature [8,9]. According to microwave theory, material characterization can be performed using two complementary methods: resonant and non-resonant [10,11]. A general rule is to use non-resonant methods when the dielectric properties of a material over a frequency range are unknown and resonant ones for a specific set of discrete frequencies [12]. Among the resonant methods, resonators such as microstrip antennas are simple, low profile non-destructive solutions to evaluate the behavior of fabrics’ dielectric properties through temperature. These antennas are mainly a sandwich structure composed of two kinds of materials: one dielectric for the substrate and another one conductive [13,14,15]. The principle of operation is using the shift of the antenna’s resonant frequency (f_r_) as a passive sensor, due to the effect of temperature changes on the dielectric properties of the fabric substrate [4]. In electromagnetism, the absolute permittivity, often known simply as permittivity explains how a material interacts when an electric field is applied to it
ε = ε_r_ε_0_ = ε^′^ − j ε^″^(1)
where ε_0_ is the vacuum permittivity (~8.85419 × 10^−12^ F⋅m^−1^) and ε_r_ is the relative permittivity, also known as the dielectric constant, where ε^′^ is the relative permittivity and ε^″^ is the loss index. The dielectric constant accounts for the molecules’ polarization in the material, when an electric field is applied. The dielectric constant increases with temperature, due to the increased mobility of polar molecules, which allows them to align more easily with the electric field [16]. The higher the frequency of operation, the more sensitive the antenna is to temperature changes, due to the shorter wavelength. In this study, we did not consider the size variations due to thermal expansion of the antenna, since the porous nature of fabrics, makes this effect negligible [3]. In addition, we minimized the effect of moisture content, by using encapsulation inside an insulator box and limiting the time of each measurement to 15 min. Other precautions such as thermal sleeves were used during the test campaigns. 

In addition to dielectric characterization through temperature, microstrip patches fabricated and tested within this paper can be used for passive temperature sensing within wearable applications. They provide advantages such as low cost, low profile, lightweight, integration into clothing and shielding effect of/from the body due to the full ground plane [17,18]. Within wearable systems, flexibility and conformability are fundamental characteristics in order to include the functionality of computers into individuals’ daily lives. Allowing the user to benefit from the performance of the system without restricting the user activity and causing any behavior modification [19,20,21]. Different options for dielectric materials arise for building flexible wearable antennas, such as several types of papers, Kapton, polyethylene terephthalate (PET) [22], polydimethylsiloxane (PDMS) [23], liquid-crystal polymer (LCP) [24] and textiles [25]. Among them, fabrics withstand bending, twisting and stretching. Furthermore, they are thin and have a low dielectric constant (ε_r_) and therefore they are good candidates for flexible wearable antennas as dielectric substrates [26]. In addition to temperature sensing textile-based resonator can plays a key role in the development of a wide range of applications, such as sports analytics [27], healthcare [28], gaming [29], and emergency services [30]. 

As far as the authors are aware, there is no experimental work carried out on measuring the relationship between dielectric constant and temperature on textile materials. For the first time, a thorough test campaign was carried out and extensive results are presented. Showing a linear relationship between ε_r_ and temperature for the three frequencies analyzed and independently of the fabric substrate used. A thermal threshold has been found at 50 °C, where the system gets into a saturation status increasing the frequency deviation of the measurements. Adding a constraint to the use of this technique.

## 2. Materials and Methods

In this section, two resonant methods for measuring the dielectric properties of the fabrics are addressed. Dielectric properties and physical structure of the textiles are provided for modelling steps in further sections; also, antenna design and fabrication for the three cases of study are explained.

### 2.1. Resonant Methods

One key characteristic of resonant methods is that they are more accurate than non-resonant ones at a single frequency or several discrete frequencies. Resonant methods could be classified into resonant perturbation and resonator techniques. In the former the material perturbates, passively, a resonant cavity, while in the later the material acts as a resonator forming part of the resonant structure [10,11,12].

#### 2.1.1. Resonant Perturbation Method

In this method, the material to be measured was placed inside the cavity in an aperture and its dielectric properties were derived from the changes inside the cavity, by using conversion equations [31,32]. The changes in the cavity’s resonant frequency and quality factor were caused by the insertion of the sample. This technique has high accuracy due to the control of the cavity’s specifications and its initial conditions. As an initial step, in order to design the microstrip patch antenna, the dielectric properties of the four different fabrics considered in this study were calculated at ambient RH and temperature. For this purpose, a material characterization split cylinder (85072A, Keysight Technologies, Reading, UK), working at 10 GHz, and a material characterization SW (N1500A-003 MMS 2015, Keysight Technologies, Reading, UK) for data conversion were used. 

#### 2.1.2. Resonator Method

A resonator method consists of using a resonating structure, such as a ring or an antenna, to derive the dielectric properties from the S-parameters with a conversion technique [33]. For that purpose, a resonant microstrip patch antenna was designed with the permittivity value measured in the previous stage (Section 2.1.1.). 

This method was selected due to its simplicity, it is low-cost and low profile, and it could be easily reproduced in any research laboratory. In addition, microstrip patch antennas are intrinsically a narrow bandwidth system, which is a beneficial characteristic for sensing [11,13], because the bandwidth acts as a probe within this method. Other resonator structures, such as rings have been widely used for material characterization [34]. On the other hand, microstrip antennas are a popular solution for long-range communications, allowing them to integrate remotely the feature of passive sensing.

Furthermore, as only the antenna is under specific conditions, it would avoid damaging any expensive piece of equipment making it an ideal option for any environmental test campaign. 

After prototyping the antenna and measuring its insertion losses (S_11_) the actual value of ε_r_ was derived based on the shift of the resonant frequency [10]. This method has good accuracy due to the narrow bandwidth nature of microstrip patch antennas, where a small variation on f_r_ is easier to recognise and to measure than for other antenna structures.

Dimensions of a microstrip patch antenna are calculated using following Equations (2)–(4) [13,35,36]
W_ant_ = (c/(2f_r_)) × (√(2/(ε_r_ + 1))) and L_ant_ = (c/(2f_r_ × (√(ε_r,eff_))) − 2∆L(2)
where W_ant_ is the width of the radiation patch, c is the speed of light in vacuum. L_ant_ is the physical length, ε_r,eff_ is the effective permittivity of the substrate and ΔL is the additional line length because of fringing fields, which could be calculated from
ε_r,eff_ = ((ε_r_ + 1)/2) + ((ε_r_ − 1)/2) + (1 + (12S_ubsh_)/W_ant_)^−1/2^(3)
∆L = 0.412S_ubsh_ × ((ε_r,eff_ + 0.3)/(ε_r,eff_ − 0.258)) × ((W_ant_/S_ubsh_ + 0.264)/(W_ant_/S_ubsh_ + 0.8))(4)
where S_ubsh_ is the thickness of the fabric substrate. 

As illustrated above, the resonant frequency of a microstrip antenna is sensitive to dielectric constant variations and according to Equations (2)–(4), if ε_r_ increases the resonant frequency of the antenna decreases. The theory of this research relays on previous studies [4,37], showing that the dielectric constant of materials increases with temperature. It was shown that the dielectric constant of a piece of Terylene film increased by 0.04 for a 20 °C temperature increase, from 20 °C to 40 °C.

### 2.2. Materials Characterization

Four of the most used fabrics (organic cotton, jeans, viscose and lycra) were tested in order to take into consideration a broad selection of daily use textiles (Figure 1a–d) [38,39]. In the pictures, the different porosity of the fabrics can be seen; these air voids have an impact on the variation of the effective permittivity. Some studies have shown that there is a linear variation of the relative permittivity depending on the infill percentage [40]. An increase in the density of porous within the fabric substrate implies more air voids trapped. The dielectric constant of the air (ε_r_ = 1) is lower than the fabrics, lowering the total dielectric constant and increasing the dissipation factor value (tan δ). The dissipation factor (DF) (often known as loss tangent, tan δ) is a ratio of the loss index (ε″) and the relative permittivity (ε′)
DF = tan δ = ε″/ε′ = 1/Q(5)
where Q is the quality factor and it describes how underdamped a resonator is.

From the microscope images (Figure 1a–d), we extracted the morphology of the four textile substrates. The first three fabrics (organic cotton, jeans and viscose) were woven fabrics with multiple fibres crossing each other at different angles to form the grain, while the last one (lycra) was knitted, it was made up of a single yarn, looped continuously to produce a braided look. The two major fabric types were considered in this investigation.

For the second part of the research, we will focus on using organic cotton due to its presence in daily clothing. The values of dielectric constant and dissipation factor for the four textile substrates measured using the resonant method of cavity perturbation are given in Table 1 below.

### 2.3. Antenna Design and Fabrication

The antenna design falls into a modelled low-profile microstrip patch antenna that was slightly adapted for each of the three frequencies of operation considered in this case of study: 2.45 GHz (A), 9.45 GHz (B) and 38 GHz (C).

#### 2.3.1. Case A at 2.45 GHz

The first case was an inset feed microstrip patch antenna (Figure 2a), designed to operate around 2.45 GHz (industrial, scientific and medical band, ISM) using the Equations (2)–(4). Figure 2a represents the layout of the proposed antenna for the four textile substrates used during the numerical analysis. The corresponding parameters dimensions in mm are illustrated in Figure 2b.

Regarding materials, adhesive copper tape (with conductivity σ = 5.8 × 10^7^ S⋅m^−1^) was used to construct both the radiation patch and the full ground plane. In order to minimize errors both pieces were cut using a Graphtec Craft-Robo CC300 from Materials Engineering Laboratory at Queen Mary University of London (QMUL). These two pieces were manually attached with a thin layer of acrylic pressure-sensitive conductive adhesive with good heat resistance to the low-cost textile substrates. The impact of the thin layer of adhesive can be ignored for the purpose of this study. Finally, a standard SMA 50 Ohm (Ω) connector was soldered to both, the edge-fed of the antenna and the background plane. Final prototypes are depicted below (Figure 3a–e).

#### 2.3.2. Case B at 9.5 GHz

The second case of study was carried out in order to increase the sensitivity of the system. We kept the same simple approach of using a microstrip patch antenna. In this case, a microstrip feed was designed and fabricated to work at higher frequencies, around 9.5 GHz (Figure 3f) [13,41]. We moved the operational frequency band from the S-band (ISM, 2.45 GHz) towards the X-band (8–12 GHz). Same Equations (2)–(4) were used for designing the antenna, in which the antenna patch dimensions are 13.8 mm × 9.8 mm (width × length). These reduced dimensions imply resonating at higher in frequencies, due to the fact that the antenna size is related with the wavelength (λ) of operation, which in turn is inversely proportional to the resonant frequency
λ = c/f_r_(6)

For fabrication, the same process and techniques were followed and as substrate material, the same organic cotton was used. To interface with the vector network analyser (VNA) during the test campaigns the same connector was soldered. 

#### 2.3.3. Case C at 38 GHz

For the final case, a new antenna was designed and fabricated, following the same rationale of previous Section 2.3.2. The design proposed in [42] has been used for this frequency, optimizing the dimensions of the radiating square patch 2.7 mm × 2.8 mm (width × length) and the position of the stub through the feeding line (Figure 3g). An SSMA 2.92 mm edge-launch connector working at 38 GHz was soldered. This connector mates with the popular SMA connections from most of the laboratory pieces of equipment. In addition, this fabric (organic cotton) antenna was designed and fabricated to work at millimeter-waves (mmW), with the potential to be used within the range of the emerging 5G technology.

## 3. Results and Discussions

### 3.1. Numerical Analysis

For numerical analysis of the antenna, CST Studio Suite [43] was used to evaluate the time-domain characteristics of the antenna structure at the three different frequencies. The time-solver calculates the development of the electromagnetic fields through time at certain spatial spots and at discrete-time samples, using Maxwell’s equations [44]. 

The antenna’s performance was analyzed both in off body and on a body phantom. The phantom model consisted of a 44 mm thick four-layer block. The phantom was modeled as 1 mm of skin, 3 mm of fat, and 40 mm of muscle. The antenna under test (AUT) was placed on top of the four-layer block, leaving a 1 mm air gap in between (see Figure 4a,b). The dielectric properties and conductivity of the three different tissues have been obtained from [45] and are listed in Table 2.

#### 3.1.1. Case A at 2.45 GHz

The reflection coefficient describes how much of an electromagnetic wave is reflected due to a discontinuity in the transmission medium, it is often known as return loss or simply S_11_. A comparison of the simulated S_11_ in off-body versus on-body is shown in (Figure 5a). As for insertion losses, the presence of a human body barely perturbed the antenna’s performance. This is an expected result due to the use of a full ground plane, which isolates the antenna from the body. In fact, this is one of the reasons of choosing a microstrip patch model with a full ground plane, for wearable applications. In addition, the ground plane helped to focus the antenna’s radiation on the broadside. The directivity (D) is a parameter that quantifies this ability to focus the radiation from the antenna. For this case, the microstrip patch had a computed directivity of 7.44 dBi towards the expected direction of propagation, and its realized gain (G) was 3.81 dB with an efficiency (eff) 43.4%
G = D × eff(7)
Simulation results are shown, in 3D, in Figure 5b.

#### 3.1.2. Case B (9.5 GHz) and Case C (38 GHz)

For the next two frequencies cases: the case B at 9.5 GHz and the case C at 38 GHz, the analysis was focused on verifying the resonance frequency and the antenna behavior under general conditions. Since the impact of human body can be neglected, as shown before, we have omitted it for these two cases. The return loss for both frequencies and results are shown in (Figure 6a,b), respectively.

### 3.2. Experimental Setup and Results

A test campaign was carried out at the Antennas Measurement Laboratory facilities of QMUL. To examine radiation patterns at far-field distances, the AUT was placed inside the anechoic chamber (Figure 7a). A diagram of the actual experimental setup used for characterizing the dielectric properties through temperature is depicted in Figure 7b. 

The prototypes were placed inside a mobile antenna electromagnetic compatibility (EMC) screened anechoic chamber to examine the radiation patterns of the antenna under test (Figure 7a). The EMC chamber was equipped with two open boundary quad-ridge horn antennas (probe) operating from 400 MHz to 6 GHz (3164-06, ETS-Lindgren, TX, USA) and from 0.8 to 12 GHz (QH800, Satimo, Haydock-MSY, UK), allowing vertical and horizontal linear polarization measurements.

The AUT was located on top of a hot plate (RCT Basic, IKA, Oxford, UK) and a thermocouple as close as possible without interfering to measure hot plate’s temperature (Figure 7b). Data from the thermocouple were correlated with the hot plate’s internal thermometer to verify the antenna’s temperature. To ensure the temperature stability and repeatability in the measurements, a due time of 15 min was allowed and 10 measurements (every ten seconds) were taken for each one of the temperatures. With these waiting periods, we guaranteed that the antenna was under the desired temperature in each step. In order to reduce the possible effects of external factors, and in particular of relative humidity (RH) variations, the setup was placed inside an insulator box made of foam. Coaxial cables close to the hot plate were protected with thermal insulator sleeves to avoid any damage to the equipment used and to minimize the impact on the measured magnitudes. 

Return losses were measured with vector network analyzers (VNA) (Keysight Technologies, Reading, UK). A PNA-L N5230C for cases A and B, and PNA-X 5244A for case C. The VNA was calibrated at the end of the coaxial cable with an E-cal kit to suppress the effects of cables and connectors, and to have the same initial reference for all our measurements.

#### 3.2.1. Case A at 2.45 GHz 

First, a general test campaign for the organic cotton model at ISM frequency was carried out in order to evaluate the overall performance of the antenna’s design. Comparison analysis between numerical and experimental performance was carried for the off body scenario.

Figure 8a shows the computed reflection coefficient of the textile antenna versus the fabricated prototype. The numerical estimation and experimental values of the prototype show a good agreement. The resonant frequency was slightly shifted (45 MHz) towards lower frequencies, due to fabrication tolerances. 

The radiation pattern properties in an off-body environment were measured in an anechoic chamber at the QMUL antennas laboratory, showing an expected behavior of a standard high Q-factor microstrip patch antenna. The E-plane cut shows that measurements match simulations fairly well, in terms of radiation pattern and directivity (Figure 8b), and behave as expected from a directional microstrip patch antenna.

##### Thermal Characterization at 2.45 GHz

The initial thermal characterization using the test setup exemplified in (Figure 7b) was performed for the woven and knitted textiles (four initial fabric substrates: cotton, jeans, viscose and lycra). 

The thermal test campaign consists of taking ten measurements of the resonant frequency for each temperature step (20 °C, 30 °C, 40 °C, 50 °C to 60 °C). The average frequency shifts (in MHz) of each fabric are listed in Table 3. Results of all measurements are depicted in the graphs below, for organic cotton (Figure 9a), jeans cotton (Figure 9b), viscose (Figure 9c) and lycra (Figure 9d).

From the test campaign, a linear behavior between ε_r_ and temperature was observed. An average shift of 10 MHz per 10 °C increment was measured for all four textiles substrates up to a temperature of 50 °C, which according to Equations (2)–(4) is equivalent to approximately a 1.67 × 10^−2^ change in the dielectric constant for each step. All results are shown in the first row of the first column of Table 4 and Table 5. For all four textiles, there was a 10% of frequency deviation (1 MHz) and thermal threshold at 60 °C where the resonant frequency tended to saturate. At the thermal threshold, the standard deviation of the resonant frequency showed a larger standard deviation, as well.

#### 3.2.2. Case B at 9.5 GHz 

The same thermal measurements were done for the second antenna, case B (Figure 10a). In this case, a 40 MHz decrement per 10 °C increase was measured. Following the same mathematical approach as in 3.2.1., the equivalent change in the dielectric constant is equal to 1.67 × 10^−2^. The result matches the one from the previous case A independently on the final resonant frequency. In this case the frequency deviation is 2.5 MHz (6.25%). Final results are summarized in the first row of the second column of both tables, Table 4 and Table 5. 

We performed a finer temperature sweep from 20 °C to 40 °C using 5 °C steps. A 20 MHz shift for each step were measured (Figure 10b), half from the 10 °C case, with a variation of 8.35 × 10^−3^ (ε_r_). These results show that the relative change of the resonant frequency with temperature had a clear linear behavior. The set of results are listed in the second row of the second column of Table 4 and Table 5.

#### 3.2.3. Case C at 38 GHz

Finally, for the third case, the same procedure as in the previous ones was used. First, measurements from 20 to 60 °C in steps of 10 °C and second from 20 to 40 °C with increments of 5 °C (Figure 11a,b) were taken. Shifts of 150 MHz and 75 MHz respectively were measured, corresponding to ∆ε_r_ of 1.67 × 10^−2^ and 8.35 × 10^−3^, with a frequency uncertainty of 5 MHz (3.33%) for this scenario. The increase in frequency shifts allowed a finer temperature sweep. In this case, an extra measurement was added to cover the ambient temperature range from 20 °C to 24 °C, with a 1 °C steps (Figure 11b). The resonance change measured for each step was of 15 MHz, ∆ε_r_ of 1.67 × 10^−3^.

Results for case C are in good agreement with previous cases, A and B, showing a linear behavior as expected, independently of the resonant f_r_. The mmW prototype improves the sensitivity in an order of magnitude, up to 1 degree Celsius. It can be seen that increasing the sensing frequency, increases the frequency deviation in the measurement.

All the quantitative results for both frequency and dielectric constant are shown in the third column of Table 4 and Table 5.

## 4. Conclusions

This paper demonstrated the efficiency and simplicity of the resonator method to accurately characterize flexible substrates, such as textiles, under environmental conditions. It also shows the cost-efficiency of the technique proposed. This enables a remote sensing scheme for services within harsh environments where equipment can be damaged or it cannot be placed.

As far as the author’s knowledge goes, this paper presents for the first time measured results of dielectric properties variation of fabrics over temperature. First, different fabric substrates (cotton, jeans, viscose and lycra) were measured at 2.45 GHz over a temperature range from 20 to 60 °C, at 10 °C steps. As no essential difference was observed among the four textiles, a finer temperature characterization was carried out to focus on organic cotton. Temperature steps were reduced from 10 °C to 5 °C at 9.5 GHz and to 5 °C/1 °C at 38 GHz respectively. 

From the test campaigns, it was observed that a linear relationship between the change in temperature and the change in dielectric constant exists and it is frequency independent. It was quantified to be ∆ε_r_ 1.67 × 10^−3^ per degree Celsius. This relation can be linearly extrapolated to any temperature value. For all the cases, within the four substrates and at the three different frequencies a saturation status behavior can be seen when heating up the substrate above 50 °C. Limiting the use of this technique up to that temperature range. 

The textile antenna working at mmW (38 GHz) presents a substantial potential as a passive temperature sensor. Several applications such as food logistic or on-body sensing could benefit from its sensitivity up to one degree Celsius and its characteristics of a fabric-based device.

For future work would be to improve the test setup to remove some uncertainties in the measurements caused by environmental factors. See the impact of going even higher in frequency in terms of impact on physical properties. Looking into the thermal threshold, actual value and plausible cause, like rarefaction of the air trapped within the resonator structure.

## Figures and Tables

**Figure 1 materials-13-01271-f001:**
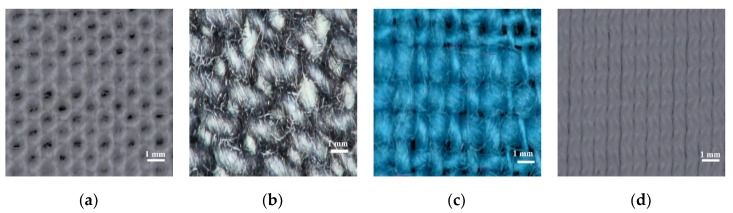
Microscope images for textiles substrates: (**a**) Cotton; (**b**) Jeans; (**c**) Viscose and (**d**) Lycra.

**Figure 2 materials-13-01271-f002:**
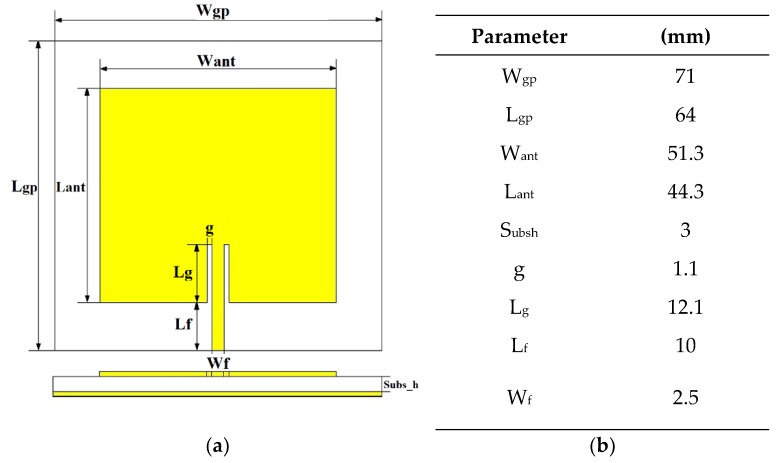
(**a**) Modelled textile antenna (**b**) Antenna parameter dimensions.

**Figure 3 materials-13-01271-f003:**
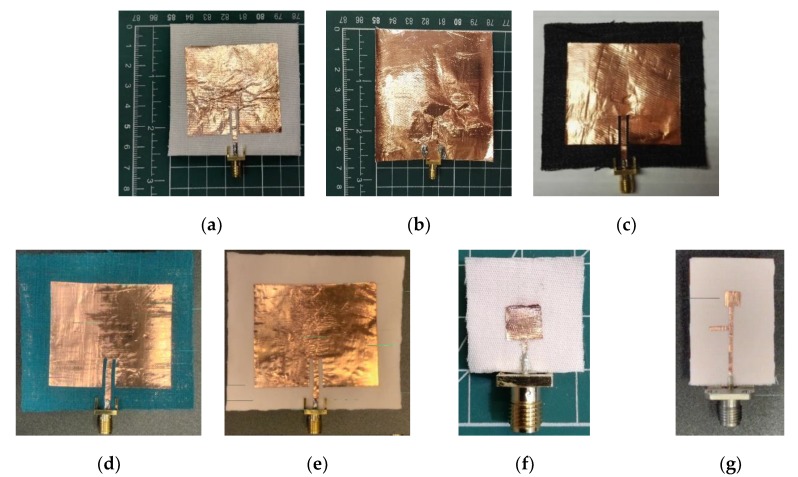
(**a**) Cotton top-view; (**b**) Cotton bottom-view; (**c**) Jeans; (**d**) Viscose; (**e**) Lycra; (**f**) Cotton Case B—9.45 GHz and (**g**) Cotton Case C—38 GHz.

**Figure 4 materials-13-01271-f004:**
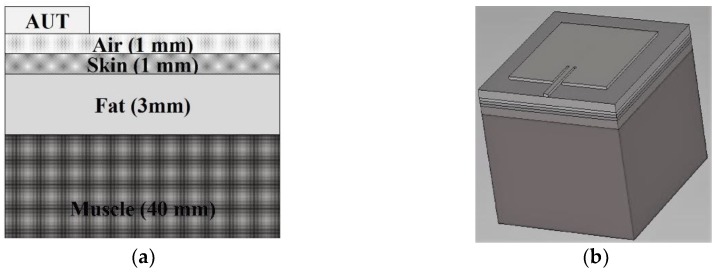
(**a**) 2D Body phantom model of four layers and (**b**) 3D body phantom model of four layers.

**Figure 5 materials-13-01271-f005:**
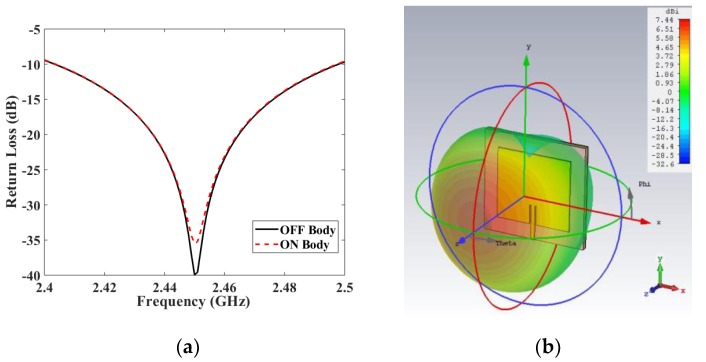
(**a**) S_11_ simulated in both off and on body and (**b**) antenna directivity (dBi) in spherical 3D coordinates.

**Figure 6 materials-13-01271-f006:**
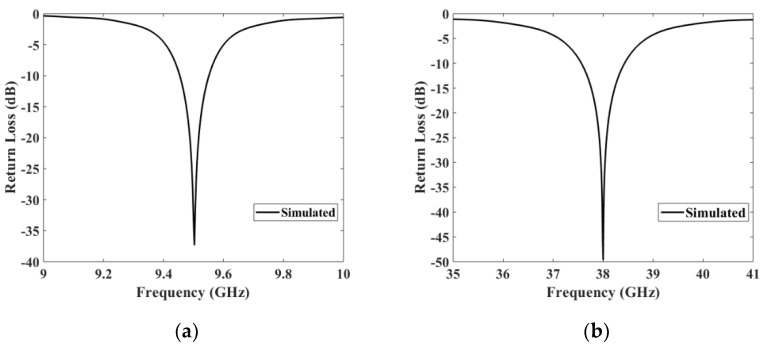
Return losses simulated for (**a**) S_11_ at 9.5 GHz and (**b**) S_11_ at 38 GHz.

**Figure 7 materials-13-01271-f007:**
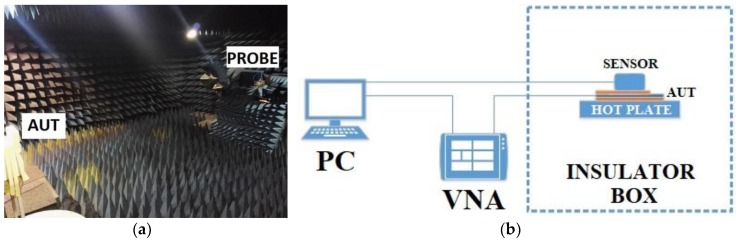
(**a**) Electromagnetic compatibility (EMC) anechoic chamber at Queen Mary University of London (QMUL) and (**b**) Measurement setup diagram.

**Figure 8 materials-13-01271-f008:**
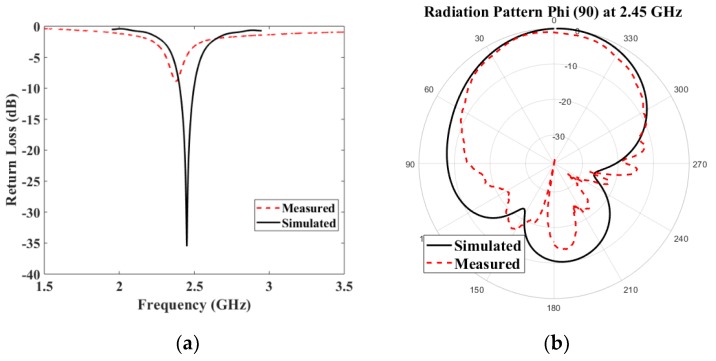
Antenna on cotton fabric at 2.45 GHz; (**a**) reflection coefficient S_11_ computed vs measured and (**b**) radiation pattern Phi (ϕ) 90 computed vs measured.

**Figure 9 materials-13-01271-f009:**
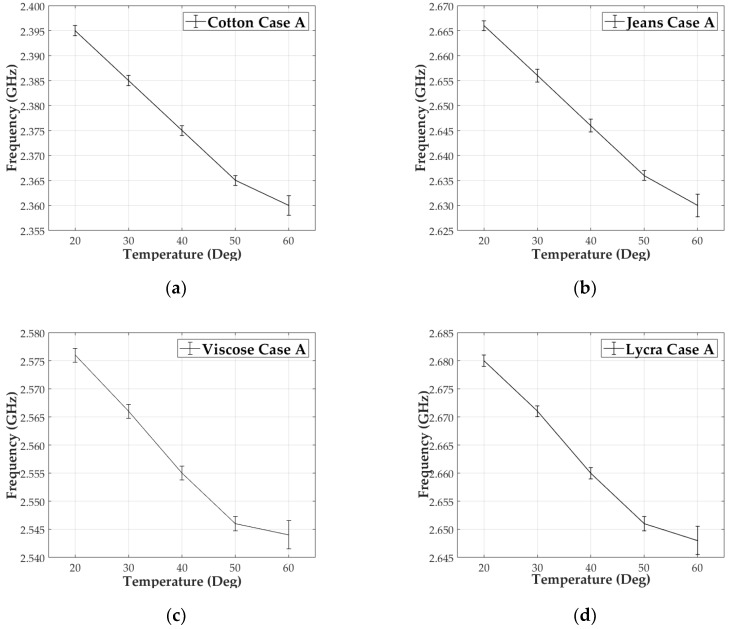
Measured frequency vs temperature representation. Case A at 2.45 GHz: 20–60 °C/10 °C steps for: (**a**) Cotton; (**b**) Jeans; (**c**) Viscose and (**d**) Lycra.

**Figure 10 materials-13-01271-f010:**
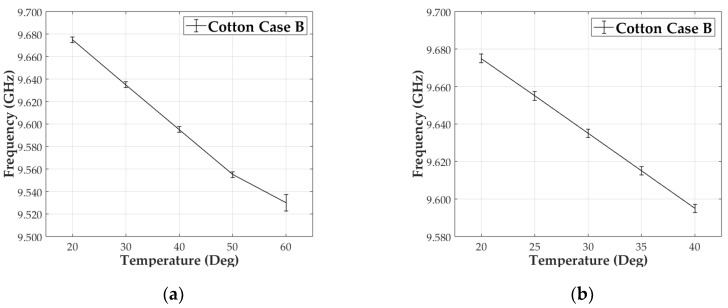
Measured frequency vs temperature representation. Cotton Case B at 9.45 GHz for: (**a**) 20–60 °C/10 °C steps and (**b**) 20–40 °C/5 °C steps.

**Figure 11 materials-13-01271-f011:**
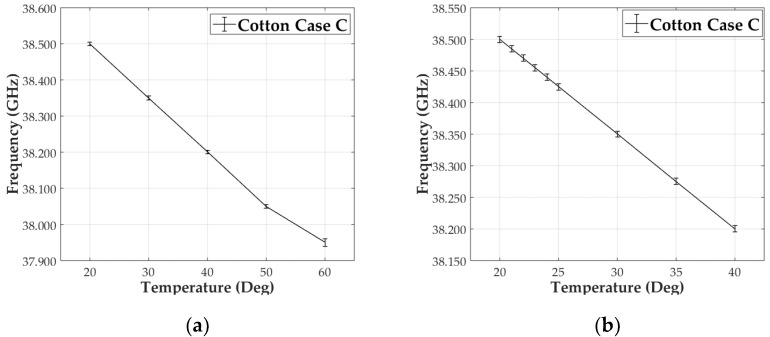
Measured frequency vs temperature representation. Cotton Case C at 38 GHz for: (**a**) 20–60 °C/10 °C steps and (**b**) 20–40 °C/5 °C steps and 20–24 °C/1 °C steps.

**Table 1 materials-13-01271-t001:** Dielectric properties of fabrics.

Fabric	ε_r_	tan δ
Cotton	1.58	0.02
Jeans	1.62	0.018
Viscose	1.64	0.016
Lycra	1.68	0.008

**Table 2 materials-13-01271-t002:** Dielectric properties of human tissues at 2.45 GHz [45].

Tissue	ε_r_	tan δ	σ (S/m)
Dry skin	38.007	0.28262	1.464
Fat	5.2801	0.14524	0.1045
Muscle	52.729	0.24194	1.7388

**Table 3 materials-13-01271-t003:** Frequency shift over temperature sweep (20–60 °C per 10 °C) at 2.45 GHz (measured results).

Temperature Increment	Cotton	Jeans	Viscose	Lycra
10 Deg	9/11 MHz	9/11 MHz	9/11 MHz	9/11 MHz

**Table 4 materials-13-01271-t004:** Frequency shift over temperature sweep (measured results in MHz).

Temperature Increment	2.45 GHz	9.5 GHz	38 GHz
10 Deg	10 MHz	40 MHz	150 MHz
5 Deg	N/A	20 MHz	75 MHz
1 Deg	N/A	N/A	15 MHz

**Table 5 materials-13-01271-t005:** Dielectric constant change over temperature sweep (measured results Δε_r_).

Temperature Increment	2.45 GHz	9.5 GHz	38 GHz
10 Deg	1.67 × 10^−2^	1.67 × 10^−2^	1.67 × 10^−2^
5 Deg	N/A	8.35 × 10^−3^	8.35 × 10^−3^
1 Deg	N/A	N/A	1.67 × 10^−3^

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
