# Peer review of "Dielectric Characterization of Non-Conductive Fabrics for Temperature Sensing through Resonating Antenna Structures"

_materials, 2020, doi:10.3390/ma13061271_

Round 1

Reviewer 1 Report

  • I prefer keywords in alphabetical order
  • Title is wierd, electromagnetic behavior of non-cunductive material? It does not make sense. The same is also written in abstract. You use antenna compound from non-conductive and conductive components, then we can discuss elmg properties, not elmg properties of non-conductive fabrics.
  • the last paragraph of the introductionmentions ref [25]. It seems to be the most importnat references (only [25] is mentioned ), but are not described in deep. In fact, it is the only one which proves some novelty of the paper.
  • cotton and jeans microstrip antennas are known - first sentence of the first paragraph 2.2 should be rewritten
  • conductive ground plane for wearables is obvious
  • fig. 6b should be more visible
  • measuerment of off body is not described
  • temperature measurement method is also described in several papers
  • measurement results do not show the quantification of delta epsilon r mentionedin the conclusion
  • methods are standard ones, the novelty is not supported by references

Author Response

First, we would like to thank you for your time reviewing this paper. Your comments and suggestions helped us to improve the quality of this work.

1) We have updated the order of the keywords (alphabetically).

2) Electromagnetic behavior has been changed for "Dielectric properties", to make it more clear that the paper is referring to dielectric properties, focus on the dielectric constant of the materials.

3) We have improved the introduction by adding references for metamaterial technologies (widening the background literature).

[20, 21] We have discussed it with the authors of [Curvilinear MetaSurfaces for Surface Wave Manipulation, Scientific reports 9 (1), 3107, 2019] and we have aggregated metamaterials within the introduction. We have added another article [Metasurfaces; High Temperature Sensing with Refractory Plasmonic Metasurfaces] as well.

[28] We have used the nanoparticles paper [Electromagnetic Nanoparticles for Sensing and Medical Diagnostic Applications, Materials 11 (4), 603, 2018] as reference for explaining the rationale behind this research work.

[27] Was mentioned twice (lines 66 and 133).

[29] Reference was added. To back up the rationale behind this research.

4) We have highlighted the novelty (in the paper) and the importance of this work; lines 65-66, 88, 317-318, and adding previous new references.

5) Rewritten and added two references [36, 37].

6) We have adapted Figure 6b (bigger Figure 6b and smaller Figure 6a).

7) Lines 257-263 explain the comparison between ON and OFF the body.

8) We have added more information to explain the advantages and disadvantages, and why this method is selected.

9) We have extended the results (in the paper) sections for point 3.2.2. and 3.2.3. To improve the content of these sections.

10) Delta epsilon r is calculated from equations (2), (3) and (4) (line278). 

11) We have added in the conclusions some limitations and future work.

We attached the updated manuscript considering your points and suggestions. Thanks again for your time.

Reviewer 2 Report

The paper is clear and pretty well written. The topic is very interesting.
The following minor revisions are suggested:
  1) The introduction is really clear and all the objectives well stated.
On the other hand some recent technology developments are missing such as:

_ metasurfaces [Curvilinear MetaSurfaces for Surface Wave Manipulation, Scientific reports 9 (1), 3107, 2019]

_ metamaterials [Inverse-designed metastructures that solve equations, Science 363 (6433), 1333-1338, 2019]
_  nanoparticles [Electromagnetic Nanoparticles for Sensing and Medical Diagnostic Applications, Materials 11 (4), 603, 2018]
It would be beneficial for the reader if authors include such technologies in the introduction section to have a complete picture of the state-of-art: pro/cons should be highlighted.   2)The related method is very well detailed. Please:  2.1) Explain in details what are the advantages/disadvantages and similarities/differences of your method compared to the above-mentioned. 2.2) A deeper study of the single materials properties is needed. Such as:
2.2.1) Authors should write some lines in this paragraph to explain why they chose to use the proposed structure
2.2.2) Explore the capability of the shape and compare yours with the one contained in the above-mentioned works (introduction).
What are the advantages/disadvantages?
  3) Authors should refer to the following additional phenomena: electric/equivalent magnetic currents, surface waves and displacement currents.
How such phenomena affect the structure functionality and in particular its properties (in particular Fig.8-10)?
  4) The paper lack in application examples, such as: sensing, refractive index measurements, diagnostics, nano-therapy, nano-communications.
I would suggest to consider such applications and explaining how you can use your device for them. Please highlight what's new in yours.
A specific paragraph would be very useful for the reader.   5) Limitations and future improvements/works should be highlighted.

Author Response

First, we would like to thank you for your time reviewing this paper. Your comments and suggestions helped us to improve the quality of this work.

1) We have discussed it with the authors of [Curvilinear MetaSurfaces for Surface Wave Manipulation, Scientific reports 9 (1), 3107, 2019] and we have aggregated metamaterials within the introduction. We have added another article [Metasurfaces; High Temperature Sensing with Refractory Plasmonic Metasurfaces] as well.

We have used the nanoparticles paper [Electromagnetic Nanoparticles for Sensing and Medical Diagnostic Applications, Materials 11 (4), 603, 2018] as reference for explaining the rationale behind this research work.

2) We have added more information to explain the advantages and disadvantages, and why this method is selected.

Lines 59-61 and lines 112-118.

3) At these levels of power, the antenna’s radiation should no induce enough temperature to disturb the experiment. For real applications, the antenna/system would be calibrated.

4) Lines 52-53 with two references, highlighting the importance of temperature sensing. The main application suggested within the paper is "Body sensing" due to the characteristics of the system. It is mention in 328-330. We have added food logistics.

5) We have added in the conclusions some future work.

We attached the updated manuscript considering your points and suggestions (changes in green). Thanks again for your time.

Reviewer 3 Report

{[1]} The authors have made an interesting attempt to write a paper addressing an electromagnetic characterization of non-conductive fabrics for temperature sensing through resonating antenna structures.

{[2]} Introduction: From my point of view, introduction is short and not well focused. In a research paper, it is expected that introduction section briefly explains the starting background and, even more important, the originality (novelty) and relevancy of the study is well established. Once this is done, hypothesis and objectives of the study need to be addressed, as well as a brief justification of the conducted methodology. It is my belief that, in this case, authors do not put effort enough (or any effort) in highlighting the relevancy and (specially) the novelty of the study. Consequently, both major aspects are compromised. I strongly recommend that authors clearly explain all these aspects (including hypothesis and objectives) in order to add scientific rigor to the manuscript.

{[3]} The authors mention that “all results for both frequency and dielectric constant are shown in the third column of Table 5 and Table 6”. Please analyse the results in more detail. The result analysis, as it is, is quite brief.

{[4]} This paper seems to be fairly crafted, yet with some limitations. However, the effort is very commendable.

Author Response

First, we would like to thank you for your time reviewing this paper. Your comments and suggestions helped us to improve the quality of this work.

- We have added three references within the introduction to cover the technology development of metamaterials within the scope of this study. And we have aggregated another paper of nanoparticles to back up the rationale behind this research work.

- We have highlighted the novelty (in the paper) and the importance of this work; lines 65-66, 88, 317-318.

- We have extended the results (in the paper) sections for point 3.2.2. and 3.2.3.

Attached you could find the updated manuscript (changes in green).

Reviewer 4 Report

The manuscript “Electromagnetic Characterization of Non-conductive Fabrics for Temperature Sensing Through Resonating Antenna Structures” simulated and designed microstrip patch antennas at the resonance frequencies of 2.45 GHz on cotton, jeans, viscose and lycra, and microstrip patch antennas at frequencies of 9.45 GHz and 38 GHz on cotton. The mirostrip patch antennas were simulated for their reflection coefficient, antenna directivity, and radiation pattern. The same parameters tested in EMC anechoic chamber match the simulation results well, which means the design and simulations are successful. The patch antennas were characterized with temperature variations from 20ºC-60ºC, and it is found that 9.45 GHz and 38 GHz antennas are respectively 4 and 15 times more sensitive than that at 2.45 GHz, and the 38 GHz antenna is able to detect 1 degree temperature variations. The microstrip patch antennas are well designed and characterized, and it is with interest on wearable devices. I would like to recommend its publication on Materials.

Some minor revisions are suggested as below:

  1. In line 160, what is the glue used and what is the process of gluing, will the glue affect the antenna performance?
  2. The words in Figure 2(a) are not clear.
  3. In Table 4 caption, should “20 °C – 60 °C” be “20 °C – 60 °C per 10 °C”?

Author Response

First, we would like to thank you for your time reviewing this paper. Your comments and suggestions have helped us to improve the quality of this work.

1) In line 160, what is the glue used and what is the process of gluing, will the glue affect the antenna performance?

It is an acrylic pressure-sensitive adhesive (with good resistance to heat, oxidisation, solvents and oils) from the brand 3M. The adhesive has conductive particles, which provide low contact resistance between the substrate and the backing to drain static charge.

Due to it is a thin adhesive film, it will have minor impacts. In comparison with the rest of the structure (in terms of thickness), the adhesive layer is super thin, therefore the total dielectric constant of the substrate is barely affected. Even if it did, it would have a minor impact on the initial resonant frequency, without influencing the development of the experiment and its outcomes. Despite of being an adhesive with conductive particles, this will have an impact on the performance of the antenna in terms of efficiency (degrading it). As the parameter of interest is the resonant frequency, this effect can be neglected.

2) We have updated Figure 2(a).

3) We have updated the caption of Table 4.

We attached the updated manuscript considering your points and suggestions (changes in green).

Round 2

Reviewer 1 Report

  • I prefer keywords in alphabetical order
    • Done
  • Title is wierd, electromagnetic behavior of non-cunductive material? It does not make sense. The same is also written in abstract. You use antenna compound from non-conductive and conductive components, then we can discuss elmg properties, not elmg properties of non-conductive fabrics.
    • Done
  • the last paragraph of the introduction mentions ref [25]. It seems to be the most importnat references (only [25] is mentioned ), but are not described in deep. In fact, it is the only one which proves some novelty of the paper.
    • can be improved, it is one of the most important part of the paper
    • The key point is it should tell the reader what is the main point od the paper, and why the reader should keep reading. Therefore:
      • location of [27] does not make sense. line 66.
      • paragraphs starting: This paper covers the concept + As far as the authors are aware.. can be put together, it is your contribution.
      • In this paragraph avoid using reference, which you describe generally. They should be placed before this paragraph appears, e.g., "Previous works [27, 29] have studied the concept of temperature sensing, but without considering the experimental results on several scenarios."
  • cotton and jeans microstrip antennas are known - first sentence of the first paragraph 2.2 should be rewritten
  • conductive ground plane for wearables is obvious
  • fig. 6b should be more visible
    • I think I refered to "Measurement setup diagram and photos
      (w/o insulator box).". Nevermind, these pictures just illustrates you performed the experiments
    • I could not find any difference between 1st and 3rd version of 6b.
  • measuerment of off body is not described
    • Lines 257-263 explain the comparison between ON and OFF the body. - IT IS NOT TRUE, do I have the latest version of the paper?
  • temperature measurement method is also described in several papers
  • measurement results do not show the quantification of delta epsilon r mentionedin the conclusion
  • methods are standard ones, the novelty is not supported by references
    • You do not state, what autors conclude in their papers you refer to. An example is: "Kannan et al. [19] investigated the synthesis of chlorine‐substituted hydroxyapatites" as can be seen from the Materials paper called Chlorapatite Derived From Fish Scales. Great example is also "Previous works [27, 29] have studied the concept of temperature sensing, but without
      considering the experimental results on several scenarios."

"We have extended the results (in the paper) sections for point 3.2.2. and 3.2.3. To improve the content of these sections." NOT TRUE - please provide the last version of the paper, thanks.

Author Response

We have re-write/rearrange the introduction:

  • Highlighting previous works earlier in the introduction and mentioned four times.
  • We have combined information and added some new references.
  • Put together the novelty in just one paragraph.
  • And merging the plausible application in one last paragraph.

Regarding "fig. 6b should be more visible"

  • We have removed one figure and augmented the diagram to make it more clear.

"measuerment of off body is not described"

  • Lines 257-268 (Maybe it was no the updated manuscript, our mistake).

methods are standard ones, the novelty is not supported by references

  • The novelty does not rely on the method. The references were used in order to reinforce the suitability of the method for this research.

"We have extended the results (in the paper) sections for point 3.2.2. and 3.2.3. To improve the content of these sections." 

  • Last version attached with these answers.

Thank you again for your time and suggestions.

Round 3

Reviewer 1 Report

  • I prefer keywords in alphabetical order
    • Done
  • Title is wierd, electromagnetic behavior of non-cunductive material? It does not make sense. The same is also written in abstract. You use antenna compound from non-conductive and conductive components, then we can discuss elmg properties, not elmg properties of non-conductive fabrics.
    • Done
  • the last paragraph of the introduction mentions ref [25]. It seems to be the most importnat references (only [25] is mentioned ), but are not described in deep. In fact, it is the only one which proves some novelty of the paper.
    • You changed it, however, the key point is missing.
    • Introduction - each paragraph should keep a key information
      • Dielectric characterization...why it is necessary to study it - OK
      • According to....method + application - it can be separated
      • As far as the authors...your novelty - this should be the last paragraph of the introduction (division of the paper is not necessary), in this paragraph other regerences are not welcomed
      • "there is no experimental work carried out on measuring the
        relationship between dielectric constant and temperature on textile materials [4]" what does the reference [4] mean? Why is it here? It seems to be the most important reference in the paper, however, not many information about authors conclusion is provided. You state your are one only one, but maybe someelse is doing the same thing..[4]< what is the difference, i.e., contribution of the [4]? (impact of temperature on textiles is not sufficient)
      • The principle of operation...you returned to the idea of method description, it should be in one paragraph + you added your limits of your paper - it should be within the novelty (last paragraph of the introduction) paragraph
      • Microstrip patches fabricated...you return to the idea of application
      • Different options for dielectric...materials and application - you can added to the application paragraph
      • This paper is divided...this is obvious
  • cotton and jeans microstrip antennas are known - first sentence of the first paragraph 2.2 should be rewritten
    • Done
  • conductive ground plane for wearables is obvious
  • fig. 6b should be more visible
    • Done
  • measuerment of off body is not described
    • You use On-body term and off-body and free-space, you can use free-space also in the picture
  • temperature measurement method is also described in several papers
    • Done
  • measurement results do not show the quantification of delta epsilon r mentionedin the conclusion
    • Done
  • methods are standard ones,
    • Done - I was just tryng to find novelty
  • the novelty is not supported by references
    • Done
  • Figure 5. (a) - missing descriptions - it has picture b and c, described as a, b, and c
  • Figure 7. - it not highlighted as a new version, however, this is much better+ the description is sufficient

Author Response

Thank you for your prompt review. We attach the new version and answers for the pending points.

the last paragraph of the introduction mentions ref [25]. It seems to be the most importnat references (only [25] is mentioned ), but are not described in deep. In fact, it is the only one which proves some novelty of the paper.

  • You changed it, however, the key point is missing.
  • Introduction - each paragraph should keep a key information
    • Dielectric characterization...why it is necessary to study it - OK
    • According to....method + application - it can be separated
    • As far as the authors...your novelty - this should be the last paragraph of the introduction (division of the paper is not necessary), in this paragraph other regerences are not welcomed
    • "there is no experimental work carried out on measuring the
      relationship between dielectric constant and temperature on textile materials [4]" what does the reference [4] mean? Why is it here? It seems to be the most important reference in the paper, however, not many information about authors conclusion is provided. You state your are one only one, but maybe someelse is doing the same thing..[4]< what is the difference, i.e., contribution of the [4]? (impact of temperature on textiles is not sufficient)
    • The principle of operation...you returned to the idea of method description, it should be in one paragraph + you added your limits of your paper - it should be within the novelty (last paragraph of the introduction) paragraph
    • Microstrip patches fabricated...you return to the idea of application
    • Different options for dielectric...materials and application - you can added to the application paragraph
    • This paper is divided...this is obvious

Regarding your comments and suggestions for the introduction we have:

  • Rearrange the order of the introduction: we have kept first, 1) Importance of Dielectric Properties and add the significance of temperature. 2) Methods and the principle of operation of the chosen one, we have mentioned the limitations and how we tried to overcome them (lines 56-60). 3) Plausible applications separated (as you recommended) and finally the last paragraph with the contributions of this paper.
  • Removed the last paragraph (This paper is divided...).
  • Novelty and limits in the last paragraph of the introduction. Removed references from this paragraph.

conductive ground plane for wearables is obvious

  • You are 100% right. It is mentioned for people no familiar with this model.

measuerment of off body is not described

You use On-body term and off-body and free-space, you can use free-space also in the picture

  • All terms change to off-body, avoiding misunderstandings.

Figure 5. (a) - missing descriptions - it has picture b and c, described as a, b, and c

  • Thank you for pointing this out. Figure 5a description means Figure 8a, so it is a typo. We have updated the captions.

We have added in the conclusions: the independent behavior of the relationship (regarding textile type and frequency) and deeper information regarding limitations and future work.

Thank you again for your time spent reviewing this manuscript.
